



# Water enhances the formation of fragmentation products via the cross-reactions of RO₂ and HO₂ in the photooxidation of isoprene

Jiayun Xu, Zhongming Chen, Xuan Qin, and Ping Dong

State Key Laboratory of Environmental Simulation and Pollution Control, College of Environmental Sciences and Engineering, Peking University, Beijing 100871, China

*Correspondence to*: Zhongming Chen (zmchen@pku.edu.cn)

**Abstract.** The photooxidation of isoprene contributes significantly to the peroxy radical ($RO_2$) pool in the atmosphere. With a widespread decreasing trend of nitrogen oxides ($NO_x$) emissions, the cross-reactions of isoprene-derived $RO_2$ with hydroperoxy radicals ($HO_2$) become increasingly important. Yet large uncertainties remain in the effect of water vapor on the products yields in these reactions. In the present study, we investigated the photooxidation of isoprene under 30 % relative humidity (low) and 80 % RH (high) in which the cross-reactions of $RO_2$ and $HO_2$ were dominated. The experiments were conducted with ozone ($O_3$) photolysis as the hydroxy radical (OH) source. We found that in the first-generation reactions, the branching ratios for methacrolein (MACR) and methyl vinyl ketone (MVK) in the cross-reactions of β-isoprene hydroxy peroxy radicals (β-ISOPOO) and $HO_2$ under 30 % RH and 80 % RH increased to approximately three times and five times of those under dry conditions owing to a water-induced change in the complexation patterns of β-ISOPOO and $HO_2$. Based on the branching ratios achieved in this study, we estimated that the MACR and MVK emissions are enhanced by 4.7−12 and 18−34 Tg yr$^{-1}$ while the β-isoprene hydroxy hydroperoxide (β-ISOPOOH) emission is suppressed by 39−78 Tg yr$^{-1}$ on a global scale when considering the water effect. In the multi-generation reactions, the yields of formic acid (FA) and acetic acid (AA) with water vapor were raised by over fivefold than we expected, able to narrow the bias between the modeled and observed global FA productions by 20 %. Since β-ISOPOOH and MACR, as well as FA and AA, play pivotal roles in aerosol formation and growth, a better interpretation of their yields helps understand the fate of isoprene in the atmosphere and improve the accuracy of the simulations of isoprene-derived SOA burdens and chemical compositions.

## 1 Introduction

The atmosphere is abundant in free radicals including hydroxyl radicals (OH), hydroperoxy radicals ($HO_2$), and organic peroxy radicals ($RO_2$). $RO_2$ plays a central role in tropospheric atmospheric chemistry as a maintainer of atmospheric oxidative capacity and intermediate of ozone ($O_3$) and secondary organic aerosol (SOA) formation (Atkinson and Arey, 2003). Isoprene, with biogenic emissions in the majority and anthropogenic emissions from industry- and traffic-related origins and urban green spaces in the minority (Sharkey, 1996; Borbon et al., 2001), dominates the global budget of non-methane volatile organic compounds (NMVOCs) emissions (Guenther et al., 2012; Sindelarova et al., 2014; Ma et al., 2022). In the ambient atmosphere,



OH rapidly converts isoprene into isoprene hydroxy peroxy radical (ISOPOO) isomers (Atkinson et al., 2006; Wennberg et al., 2018). Further photooxidation leads to the fragmentation of the carbon skeleton and the propagation of a variety of $RO_2$. As a VOC with large emissions and high reactivity, isoprene serves as a major contributor to $RO_2$ in the atmosphere, therefore has profound implications for regional air quality and global climate (Kanakidou et al., 2005; Squire et al., 2015).

The fate of isoprene-derived $RO_2$ in the ambient atmosphere is mainly controlled by four reaction pathways, namely the NO
pathway, the $HO_2$ pathway, the $RO_2$ pathway, and the intramolecular H-shift pathway (Orlando and Tyndall, 2012; Jenkin et al., 2019). Their distributions are regulated by nitrogen oxides ($NO_x$) and hydrogen oxide radicals ($HO_x = OH + HO_2$) concentrations in a certain region (Bates and Jacob, 2019). The $HO_2$ pathway in isoprene photooxidation is a substantial source of organic peroxides in the ambient atmosphere, especially where biogenic emissions are dominated such as in Amazon (Khan et al., 2015b). On a global scale, 40−50 % of ISOPOO is consumed via the $HO_2$ pathway (Crounse et al., 2011; Bates and
Jacob, 2019), with the decreasing trend of $NO_x$ emissions even promoting its distribution. β-isoprene hydroxy peroxy radical (β-ISOPOO) isomers consist of more than 90 % of the ISOPOO radical pool under atmospheric relevant conditions (Teng et al., 2017). Generally, the reactions of β-ISOPOO isomers with $HO_2$ process through two channels: a major one produces β-isoprene hydroxy hydroperoxide (β-ISOPOOH) and a minor one produces methacrolein (MACR) or methyl vinyl ketone (MVK) and formaldehyde (HCHO) (Paulot et al., 2009; Liu et al., 2013), namely a non-fragmentation channel and a
fragmentation channel. The regions with large isoprene emissions coincide with the humid regions (Porter et al., 2021). However, given that the branching ratios for the two channels were obtained under dry conditions in the previous studies, their values remain unclear when water vapor exists (Berndt, 2012; Liu et al., 2013; Rivera-Rios et al., 2014). A computational study pointed out that water vapor can complex with ISOPOO isomers at different relative abundances, consequently affecting the branching ratios in the subsequent reactions and changing the distribution of oxidation products (Clark et al., 2010).
Moreover, for the further photooxidation of MACR, MVK, and β-ISOPOOH, whether water vapor can promote or suppress the formation of certain products? The relevant researches are more limited (Bates et al., 2014; Praske et al., 2015; St Clair et al., 2016). Accordingly, it is in urgent need to shed light on isoprene first- and multi-generation photooxidation regimes in the presence of water vapor.

In the present study, experiments were performed in an oxidative flow reactor (OFR) to investigate the photooxidation regimes
of isoprene via the $HO_2$ pathway under a wide range of OH exposure ($OH_{exp}$) in the presence of water vapor. We found that the branching ratios for MACR and MVK increased in β-ISOPOO + $HO_2$ reactions. We attributed this phenomenon to a change in the structures of the reaction intermediates when water vapor exists. In the multi-generation reactions, the yields of formic acid (FA) and acetic acid (AA) increased significantly compared to those obtained without water vapor, indicating that water-involved chemical processes led to their formation.



## 2 Experimental

### 2.1 Apparatus and procedures

Typically, the simulations of atmospheric chemical processes are conducted in two types of enclosures: environmental chambers (ECs) and oxidative flow reactors (OFRs). OFRs equipped with 254 nm UV lights applying the $O_3$ photolysis in the presence of $H_2O$ as the OH source have been widely adopted to investigate multi-generation gas-phase photooxidation regimes of single or mixed VOC precursors (Friedman and Farmer, 2018; Lau et al., 2021) and SOA formation from either specific VOC precursors or a certain emission source (Lambe et al., 2012; Lambe et al., 2015; Li et al., 2019). Here, OFR was chosen as the experimental enclosure on account of the following three advantages: first, the wall loss and photolysis loss of reactants and products in OFRs are lower than those in ECs (Peng et al., 2016; Brune, 2019; Krechmer et al., 2020; Peng and Jimenez, 2020); second, the higher $OH_{exp}$ provided by the photolysis of $O_3$ than conventional precursors such as $H_2O_2$ accelerates the photooxidation process and benefits the collection of the multi-generation products (Lambe et al., 2011); third, the interference of $H_2O_2$ to the detection of organic peroxides can be eliminated.

The isoprene photooxidation experiments took place in a 2 L quartz OFR (100 cm length, 5cm I.D.) with a water jacket for temperature control at $298 \pm 0.5$ K. Figure 1 shows the overview of the experimental apparatus. $O_3$ was generated by photolysis of $O_2$ in a quartz tube with adjustable 185 nm UV radiation and photolyzed under 254 nm UV radiation in the OFR in the existence of water vapor. Water vapor was generated by passing $N_2$ or $O_2$ (Beijing Haipubeifen Gas, Beijing, China, $\geq 99.999\%$) through a water bubbler. Dry air, water vapor, and isoprene (20.1 ppmv in $N_2$, the National Institute of Metrology, China) were blended in a quartz mixing ball before converging with a 0.1 standard L min$^{-1}$ (SLPM) flow of $O_3$ in $O_2$. A total flow of 2.0 SLPM mixed gas reactants (80 % $N_2$ and 20 % $O_2$) were introduced into the OFR with a maximum residence time of 61 s. The gas flow ran straight forward in the OFR given a Reynolds number of ~22, which indicated a laminar flow (Ezell et al., 2010). The OFR was operated in continuous flow mode (which means gas reactants were introduced from the inlet while products ran out from the outlet continuously) with a pre-run duration time of 2 h to reach a steady state before sample collection started. A movable stainless tube with Teflon lining was protruded into the OFR on the centerline to collect samples under a variety of $OH_{exp}$. The OFR and 254 nm UV lamps were placed in a stainless cover with a mirror reflection inner wall. The selected experimental conditions are listed in Table 1. The method for $OH_{exp}$ and atmospheric equivalent photochemical age (atmospheric EPA) determination can be found in the Supplement. We suggest experiments under lower $O_3$ concentrations (low $OH_{exp}$ experiments, Exp. 1 and 2) represent first- to second-generation photooxidation of isoprene, while those under higher $O_3$ concentrations (high $OH_{exp}$ experiments, Exp. 3 and 4) represent multi-generation photooxidation. The corresponding $O_3$ concentrations under 80 % RH were cut to ~40 % of those under 30 % RH to maintain $OH_{exp}$ in a comparable range. The consumption of isoprene via ozonolysis was less than 1 % in our experiments due to the enormous disparity in the reaction rate constants of isoprene with OH and $O_3$ at 298 K ($k_{OH} = 9.99 \times 10^{-11}$ cm$^3$ molec$^{-1}$ s$^{-1}$ and $k_{O3} = 1.28 \times 10^{-17}$ cm$^3$ molec$^{-1}$ s$^{-1}$) (Jenkin et al., 2015). Each experiment was repeated three times with two parallel samples collected each time. The OFR was rinsed with ultrapure water (18 MΩ, Millipore) and blown dry with $N_2$ after each experiment.




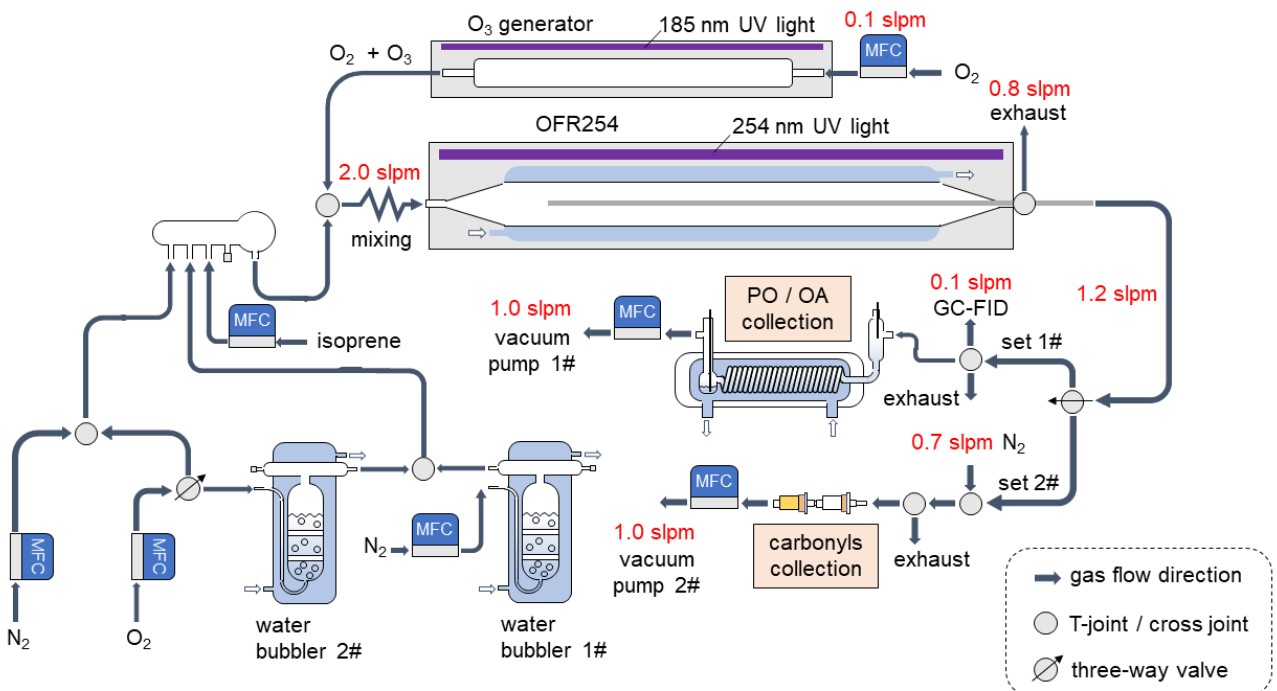

**Figure 1: Overview of the experimental apparatus.**

**Table 1: Overview of the selected experimental conditions.**

| Exp. | [ISO] / ppbv | [O₃] / ppbv | OHₑₓₚ / molec cm⁻³ s | Atmospheric EPA / hrs | RH / % |
|------|--------------|-------------|----------------------|------------------------|--------|
| 1 | $55 \pm 3$ | $407 \pm 3$ | $2.4 \times 10^9 - 3.6 \times 10^{10}$ | 0.4−6.6 | 30 |
| 2 | $55 \pm 3$ | $165 \pm 5$ | $3.3 \times 10^9 - 3.8 \times 10^{10}$ | 0.6−7.1 | 80 |
| 3 | $55 \pm 3$ | $959 \pm 16$ | $5.8 \times 10^9 - 7.3 \times 10^{10}$ | 1.1−13.6 | 30 |
| 4 | $55 \pm 3$ | $407 \pm 3$ | $7.6 \times 10^9 - 1.8 \times 10^{11}$ | 1.4−32.4 | 80 |

Note: [ISO], isoprene concentration; [O₃], O₃ concentration. The uncertainty of $OH_{exp}$ is within $\pm 15$ %.

### 2.2 Measurements of reactants and products

The concentrations of reactants (isoprene and $O_3$) and products (including peroxides, carbonyls, and organic acids) were determined using different methods. The concentration of isoprene was detected online by gas chromatography coupled with

a flame ionization detector (GC-FID, Agilent 7890A, USA) at a detection limit of 5 ppbv. The concentration of $O_3$ was determined by indigo disulphonate spectrophotometry with a detection limit of 30 ppbv. Two independent sampling modules for product collection (set 1# and 2#) can be switched through a three-way valve. The gas flow rates for sample collection controlled by mass flow controllers (MFCs) can be found in Fig. 1. In set 1#, peroxides and organic acids were collected in a glass coiling tube at $277 \pm 0.5$ K. The rinsing solution was either phosphoric acid ($H_3PO_4$) solution ($5 \times 10^{-3}$ M, pH = 3.5) for

peroxides or ultrapure water for organic acids at a rate of 0.2 mL min⁻¹. The efficiencies for products collection in the rinsing





solution were calculated based on Henry's law constants and listed in Table S1. The control experiments indicated that the aqueous phase ozonolysis of isoprene was negligible in the coiling tube. Peroxides with the number of carbon atoms no more than 2 (C≤2 PO) including $H_2O_2$, hydroxymethyl hydroperoxide (HMHP), methyl hydroperoxide (MHP), peroxyacetic acid (PAA), and performic acid (PFA) were analyzed online using high-performance liquid chromatography (HPLC-PO, Agilent

1260, USA) coupled with post-column derivatization and fluorescence detection (Hua et al., 2008). We also measured the concentration of total peroxides (TPO) with the iodometric spectrophotometric method (Li et al., 2016). Organic acids including FA and AA were analyzed using ion chromatography (IC, DIONEX ICS-2000). The detection limit for organic acids, C≤2 PO, and TPO was 100 pptv, 10 pptv, and 5 ppbv, respectively. In set 2#, 2,4-dinitrophenylhydrazine (DNPH)-silica cartridges with $O_3$ scrubbers in front were used to collect carbonyls. An additional $N_2$ flow was added before sampling to avoid

deliquescence of potassium iodide (KI) in $O_3$ scrubbers. The DNPH cartridges were eluted with 5 mL acetonitrile (Fisher, HPLC grade, ≥ 99.9 %), placed in the dark for 12 h, and analyzed by another HPLC (HPLC-CA, Agilent 1100, USA) (Wang et al., 2009). Here we measured HCHO, MACR, MVK, methylglyoxal (MGLY), and hydroxy acetone (HACE), and the detection limits for individual carbonyls varied from 15 to 60 pptv. Backgrounds of HCHO, FA, AA, and C≤2 PO were detected in blank samples, and they were subtracted from the results.

**2.3 Model simulation**

A box model equipped with the Master Chemical Mechanism version 3.3.1 (MCM v3.3.1) was applied to simulate the evolution of products under the selected experimental conditions. The explicit mechanisms for isoprene oxidation (including 1926 reactions with 602 related species) were abstracted from the MCM website (http://mcm.york.ac.uk/home.htt, last access: 18 April 2022) (Jenkin et al., 2015). The forward and backward rate constants of the $O_2$ reaction with isoprene hydroxy adducts

were updated according to Teng et al. (2017). The rate constants and branching ratios of the OH reaction with ISOPOOH isomers were modified based on St Clair et al. (2016).

**3 Results and discussion**

**3.1 Yields of first- and multi-generation products**

We calculated the $OH_{exp}$-dependent overall molar yields of the products relative to consumed isoprene, and the results are

presented in Fig. 2. Yields corrections were made to eliminate the effect of secondary OH oxidation, photolysis, and wall loss of concerning products on the determination of their actual yields (see the Supplement for detail). We define the OH conversion of isoprene into ISOPOO isomers and the ISOPOO-involved reactions as the first-generation reactions, and the corresponding closed-shell products are defined as the first-generation products. Further photooxidation transforms the first-generation products into more oxidized products. The reactions involved in this process are referred to as the multi-generation reactions,

and those products are called the multi-generation products. Certain products, such as HCHO, come from both the first- and the multi-generation reactions. We separate their formation in reactions of different generations. In our experiments, we found





the yields of MACR and MVK almost remained stable during the photooxidation process, as a typical characteristic of the first-generation products, and no significant difference in their yields between high OH$_{exp}$ and low OH$_{exp}$ experiments was noticed. The yields of MACR and MVK were 11.1 ± 1.0 % and 21.1 ± 2.6 % under 30 % RH and 17.3 ± 2.3 % and 36.4 ± 2.8 % under 80 % RH. The yields of MACR and MVK were surprisingly higher than those obtained in the experiments without water vapor (Liu et al., 2013). Berndt (2012) also reported unexpected high yields of MACR and MVK (65 % in total) in low-NO$_x$ experiments under 50 % RH, indicating a potential water effect on the β-ISOPOO chemistry.



Figure 2: OH$_{exp}$ dependent overall molar yields ($Y'_{PRO,i}(t)$) of measured products under (a−d) 30 % RH and (e−h) 80 % RH in low and high OH$_{exp}$ experiments. L, yields derived from low OH$_{exp}$ experiments (Exp. 1 and 2); H, yields obtained from high OH$_{exp}$ experiments (Exp. 3 and 4). The error bars represent ± standard deviation (± SD) based on 6 measurements.



We define organic peroxides which contain no less than three carbons as C≥3 PO, and their yields can be calculated by subtracting the yields of C≤2 PO from TPO. According to the results of the model simulation, β-ISOPOOH made up over 90% of C≥3 PO formed in low $OH_{exp}$ experiments, thus we took C≥3 PO as a surrogate for β-ISOPOOH. The yield of C≥3 PO was suppressed by water vapor in the early stage of the reactions because the formation of MACR and MVK was enhanced, whereas it began to increase later. The distribution of the $HO_2$ pathway was increasing during the photooxidation process due to an increase in $HO_2$ concentrations and a decrease in β-ISOPOO concentrations, leading to a growing yield of β-ISOPOOH. Moreover, highly oxidized multifunctional molecules (HOMs) containing at least two hydroxy peroxy groups (-OOH) are produced from the multi-generation reactions. The oxidation of β-ISOPOOH by OH produces not only the well-known product isoprene epoxydiol (IEPOX) but also isoprene dihydroxy dihydroperoxide ($ISOP(OOH)_2$) which possesses two -OOH groups (Liu et al., 2016; D'Ambro et al., 2017).

HMHP was successfully quantified in our experiments by applying $O_3$ as the OH precursor. The yield of HMHP varied with $OH_{exp}$ from ~2 % to ~1 %, independent of RH. The decreasing trend of HMHP yield with increasing $OH_{exp}$ implies that it is a first-generation product and it can lose via reactions other than photooxidation, such as the formation of peroxyhemiacetals (Ziemann, 2002). We suggest that HMHP is formed from hydroxymethyl radical ($CH_2OH$), which is a byproduct of the decomposition of β-hydroxy alkoxy radical (β-ISOPO). $CH_2OH$ reacts with $O_2$ to form hydroxymethyl peroxy radical ($HOCH_2OO$) rapidly and subsequently reacts with $HO_2$ to produce HMHP and FA (Atkinson et al., 2006). The high $HO_2$ concentration in our experiments accelerated the formation of HMHP and FA from $HOCH_2OO$.

The yields of the multi-generation products are relatively low at the beginning of the photooxidation process, while they increase with $OH_{exp}$ until the precursors are consumed completely. We found that FA, AA, MGLY, and MHP are typical multi-generation products, as their yields continuously increased during the photooxidation process. Both the first- and the multi-generation reactions contributed comparably to HCHO formation. The yields of the multi-generation products did not show evident dependence on RH except for AA, whose yield was significantly enhanced by RH, indicating the ability of water vapor to accelerate its formation.

### 3.2 Mechanisms of the first-generation reactions

To identify the relative importance of the reaction pathways of β-ISOPOO in our experiments, we calculated the reaction rates of β-ISOPOO via different pathways (units: $s^{-1}$) referred to Liu et al. (2013) considering the $HO_2$ concentrations calculated from the observed evolution of $H_2O_2$ concentrations (see Fig. S3), the modeled β-ISOPOO concentrations, and the rate constants of the $HO_2$ reactions, the $RO_2$ reactions and the intramolecular H-shift reactions of β-ISOPOO. The water complexes of isoprene-derived $RO_2$ are prevailing in the humid air (Clark et al., 2010; Khan et al., 2015a). The formation of the $RO_2$ ($HO_2$) · $H_2O$ complex enhances the reaction rate constants in the cross- and self-reactions of $HO_2$ and $RO_2$ (Alongi et al., 2006; Huang et al., 2011; Kumbhani et al., 2015). To parameterize this water effect in our calculation, we referred to an experimental study based on β-hydroxy ethyl peroxy (β-HEP), whose structure and binding energy with $H_2O$ are similar to that of β-ISOPOO





(Kumbhani et al., 2015). Details can be found in the Supplement. Compared to the reaction rate constants without water vapor,

those of the cross-reactions of $HO_2$ and β-ISOPOO are enhanced by over 20 % under 30 % RH and over 50 % under 80 % RH, while those of the cross/self-reactions of β-ISOPOO nearly doubled under 30 % RH and more than tripled under 80 % RH. Our calculation results suggested that a majority (~80 %) of β-ISOPOO was consumed via the $HO_2$ pathway while a minority (~20 %) was consumed via the $RO_2$ pathway, and the distribution of the $HO_2$ pathway increased during the photooxidation process (see Fig. S2).

Here we tentatively separated the yields of MACR and MVK from β-ISOPOO + $HO_2$ reactions by subtracting their yields via the $RO_2$ pathway, which were obtained using a box model, from the observed overall yields. It is noted that the recycled β-ISOPOO and β-ISOPO from H-abstraction at the -OOH group by OH and photolysis of β-ISOPOOH also contributed ~10 % to the formation of MACR and MVK (Fu et al., 2008; Jenkin et al., 2015), and this part was subtracted as well. The results show that the yield of MACR from the reaction of β 4-OH, 3-OO ISOPOO radical (β-4,3-ISOPOO) with $HO_2$ is 2.5 ± 1.0 %

under 30 % RH and 5.6 ± 2.8 % under 80% RH, while the yield of MVK from the reaction of β 1-OH, 2-OO ISOPOO (β-1,2-ISOPOO) with $HO_2$ is 10.2 ± 5.1 % under 30 % RH and 19.1 ± 1.3 % under 80 % RH. The yield for each β-ISOPOOH isomer was achieved by multiplying the yield of C≥3 PO and the model-simulated weight of the corresponding isomer. The yield of β 1-OH, 2-OOH ISOPOOH (β-1,2-ISOPOOH) and β 4-OH, 3-OOH ISOPOO (β-4,3-ISOPOOH) was 43.1 ± 10.6 % and 12.5 ± 1.4 % under 30 % RH, and 42.3 ± 14.9 % and 10.5 ± 1.7 % under 80 % RH.

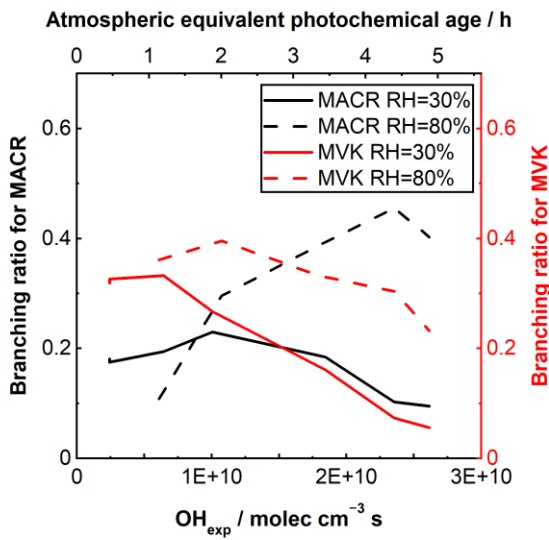


**Figure 3: OH$_{exp}$ dependent branching ratio for MACR in the reaction of β-4,3-ISOPOO with HO₂ and for MVK in the reaction of β-1,2-ISOPOO with HO₂ under 30% and 80% RH.**



The branching ratios for fragmentation in the reactions of β-1,2-ISOPOO and β-4,3-ISOPOO with $HO_2$ under 30% and 80% RH were calculated as the ratio of MVK yield to the sum of MVK and β-1,2-ISOPOOH yield, and the ratio of MACR yield to

the sum of MACR and β-4,3-ISOPOOH yield. The results are presented in Fig. 3. The branching ratio for MACR in the $HO_2$ reaction of β-4,3-ISOPOO is $0.164 \pm 0.049$ under 30 % RH and $0.330 \pm 0.122$ under 80 % RH, while that for MVK in the $HO_2$ reaction of β-1,2-ISOPOO is $0.203 \pm 0.113$ under 30 % RH and $0.325 \pm 0.055$ under 80 % RH. Combined with a branching ratio of 0.063 for MACR and MVK via the $HO_2$ pathway under dry conditions (Liu et al., 2013), we found that there is a positive correlation between the branching ratios and RHs (branching ratio = $3.27 \times 10^{-3}$ RH (%) + $7.18 \times 10^{-2}$, $R^2 = 0.99$, the

RH range is <2−80 %, the uncertainty is within ± 60 %). The linear water effect on the branching ratios indicated that $H_2O$ altered the β-ISOPOO chemistry possibly by serving as a reactant in the cross-reactions of β-ISOPOO and $HO_2$.

**Figure 4: Suggested mechanisms for β-ISOPOO + HO₂ reactions with and without water vapor.**

Theoretical studies have put forward the formation of two intermediates in the reactions of $RO_2$ with $HO_2$. A hydrogen-bonded

complex intermediate (ROO…HOO) tends to proceed to ROOH and $O_2$, whereas a hydrotetroxide intermediate (ROO…OOH) is prone to decompose into RO, OH, and $O_2$ (Hasson et al., 2005; Hou et al., 2005). Generally, for alkyl peroxy radicals, ROO…HOO is preferred, yet polar moieties such as carbonyl or hydroxy group can promote the formation of ROO…OOH via hydrogen bond. For β-ISOPOO, the β-ISOPOO…HOO channel (non-fragmentation channel) proceeds to the formation of





β-ISOPOOH, while the β-ISOPOO…OOH channel (fragmentation channel) proceeds to the formation of β-ISOPO, which

subsequently decomposes into MACR or MVK and CH$_2$OH. We suggest that the water effect on MACR and MVK branching ratios might be a result of the interactions between H$_2$O, HO$_2$, and β-ISOPOO. H$_2$O serves as "glue" here, which draws β-ISOPOO and HO$_2$ together by hydrogen bonds. Given the presence of water vapor, HO$_2$ can combine with H$_2$O effectively to form the HO$_2$·H$_2$O complex (Alongi et al., 2006). The existence of the HO$_2$·H$_2$O complex possibly decreases the energy barriers for the formation of β-ISOPOO…OOH via a hydrogen-bonded trimer complex intermediate (β-ISOPOO·H$_2$O·HO$_2$

complex, see Fig. 4), thus promoting the production of MACR and MVK. The suggested mechanisms for water-involved HO$_2$ reactions of β-ISOPOO are presented in Fig. 4.

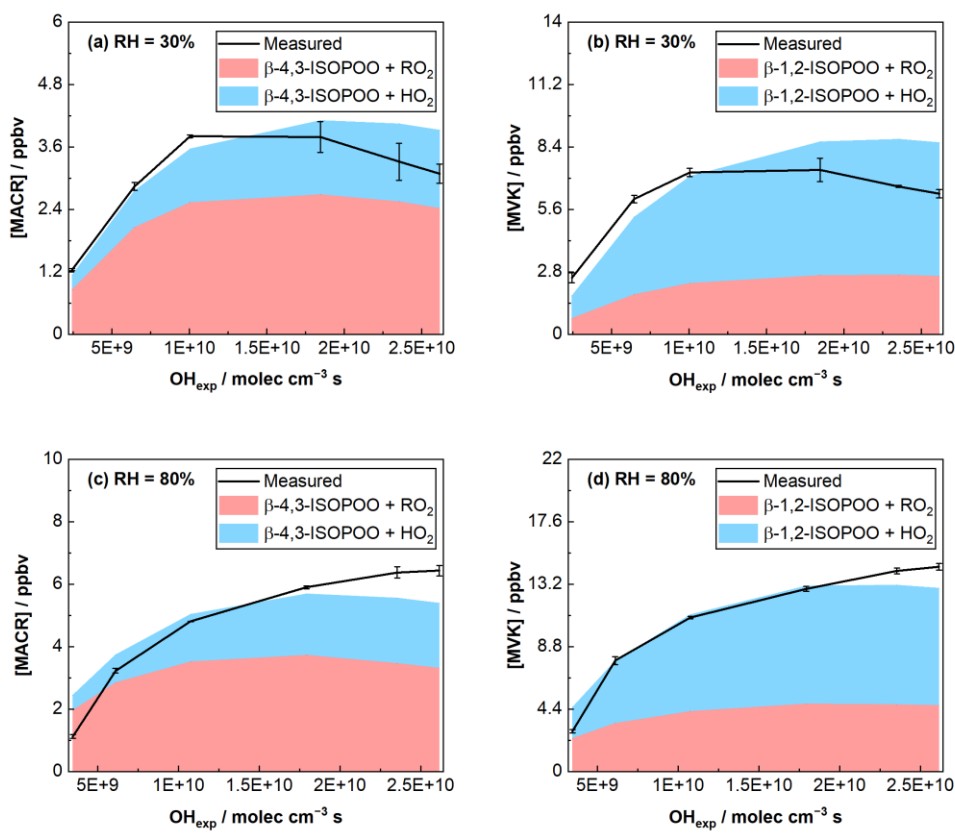

**Figure 5: Comparison of the measured and modeled evolution of MACR and MVK concentrations under (a, b) 30 % RH and (c, d) 80 % RH. Updated branching ratios for MACR and MVK in the β-ISOPOO + HO₂ reactions were involved in the modeling. Lines**

**and error bars (±SD), measured evolution of MACR or MVK concentrations; filled areas in red, modeled MACR or MVK concentrations derived from β-4,3-ISOPOO + RO₂ or β-1,2-ISOPOO + RO₂ reactions; filled areas in blue, modeled MACR or MVK concentrations derived from β-4,3-ISOPOO + HO₂ or β-1,2-ISOPOO + HO₂ reactions.**





We applied a box model constrained by the updated branching ratios for MACR and MVK in the $HO_2$ reactions of β-ISOPOO

under 30 % and 80 % RH to reproduce the evolution of their concentrations observed in our experiments, and the results were shown in Fig. 5. We found that the modeled trends met well with the observed ones. Under 30 % RH, the $HO_2$ pathway contributed 1/3 to MACR formation with the other 2/3 contributed by the $RO_2$ pathway, while the relative contributions to MVK formation by the two pathways were inverted. The relative contributions by the two pathways remained virtually unchanged under 80 % RH for MACR, whereas the contribution by the $HO_2$ pathway to MVK slightly decreased to ~60 %.

There is no denying the possibility of MACR and MVK formation from other reactions in our experiments. One possibility is

from the reaction of β-ISOPOO with $O_3$ (Orlando and Tyndall, 2012). Nevertheless, given the low rate constant for the reaction of $RO_2$ with $O_3$ (k = $1 \times 10^{-17}$ $cm^3$ $molec^{-1}$ $s^{-1}$ for the reaction of $CH_3O_2$ with $O_3$), we suggest the reaction of β-ISOPOO with $O_3$ was uncompetitive to that with $HO_2$ and $RO_2$ in our experiments. Another possibility is from the aqueous-phase decomposition of β-ISOPOOH in aerosol liquid water when the RH is high enough for SOA particles to be deliquescent (under 80 % RH in our experiments). Yet according to Fang et al. (2020), the decay of β-ISOPOOH in water seems to be too slow to

make sense, and we suggest the contribution of this reaction to MACR and MVK formation is also negligible.

To evaluate the recovery of the first-generation products, we calculated the sum yields of MACR, MVK, and β-ISOPOOH in first-generation reactions. During the photooxidation process, its value was maintained at ~82 % under 30 % RH and increased from ~75 % to >90 % under 80 % RH, respectively. The growing trend under 80 % RH was attributed to the increasing distribution of the $HO_2$ pathway (see Fig. S2). Lost carbons mainly resulted from undetected products via the $RO_2$ pathway

such as dihydroxy isoprene (Ruppert and Heinz Becker, 2000; Claeys et al., 2004; Paulot et al., 2009; Bates et al., 2021).

In essence, $CH_2OH$ is formed concurrently with MACR or MVK from the decomposition of β-ISOPO, which reacts efficiently with $O_2$ to generate $HOCH_2OO$ and subsequently decompose into HCHO and $HO_2$ (Atkinson et al., 2006). Consequently, a theoretically 1:1 ratio of the sum of MACR and MVK yield to HCHO yield in first-generation photooxidation was expected, as has been reported in several previous studies (Miyoshi et al., 1994; Jenkin et al., 1998; Benkelberg et al., 2000; Ruppert and

Heinz Becker, 2000). However, according to Fig. 6, we found this ratio significantly higher than 1:1 in our experiments, especially at the early stage under 80 % RH, indicating a large proportion of C1 products was missing. A recent theoretical study pointed out that water molecules can slow down the reaction of $CH_2OH$ with $O_2$ to generate HCHO and $HO_2$ by several orders of magnitude (Dash and Akbar Ali, 2022). We suggest that this water effect led to an increased proportion of $CH_2OH$ being processed through self-reaction to produce $CH_2OH$ dimers ($C_2H_6O_2$) (Grotheer et al., 1985). Moreover, given two orders

of magnitude higher $HO_2$ concentration in our experiments than in the ambient atmosphere, we suppose the reaction of $HOCH_2OO$ with $HO_2$ likely plays a more critical role, through which HMHP and FA are generated with a branching ratio of 0.6 and 0.4, respectively (Atkinson et al., 2006; Jenkin et al., 2007). We also proposed the possibility for the reaction of $CH_2OH$ with OH to form hydrated formaldehyde ($CH_2(OH)_2$), which can dehydrate into HCHO later and explain the decreasing trend of the ratios with $OH_{exp}$ in Fig. 6. Other reaction pathways involving $HOCH_2OO$ and $CH_2(OH)_2$ may also contribute to FA

formation (see Table S6). We assume the generation of organic aerosols in isoprene photooxidation when $OH_{exp}$ exceeded $1 \times$





$10^{10}$ molec cm$^{-3}$ s (Kroll et al., 2006; Nguyen et al., 2011; Brégonzio-Rozier et al., 2015). Considering the SOA formation, the hemiacetal reaction would occur between $CH_2(OH)_2$ and HCHO in deliquesced aerosols under 80 % RH to produce HCHO oligomers [$HO(CH_2O)_nH$] (Toda et al., 2014). Based on the observed yields of HCHO, HMHP, and FA when the atmospheric EPA was less than 1 h (when the first-generation reactions were dominated), we suggest that the yields of FA and HCHO in

the first-generation reactions are 0.061 and 0.124 under 30 % RH and 0.046 and 0.061 under 80 % RH in our experiments. 16.3 %, 4.9 %, and 38.0 % of theoretical HCHO yield in the first-generation reactions under 30 % RH (which equals the sum of MACR and MVK yield) was replaced by FA, HMHP, and undetected products such as $C_2H_6O_2$ and $CH_2(OH)_2$, while under 80 % RH the replaced proportion was 12.0 % for FA, 4.1 % for HMHP, and 70.8 % for $C_2H_6O_2$, $CH_2(OH)_2$ and HCHO oligomers.

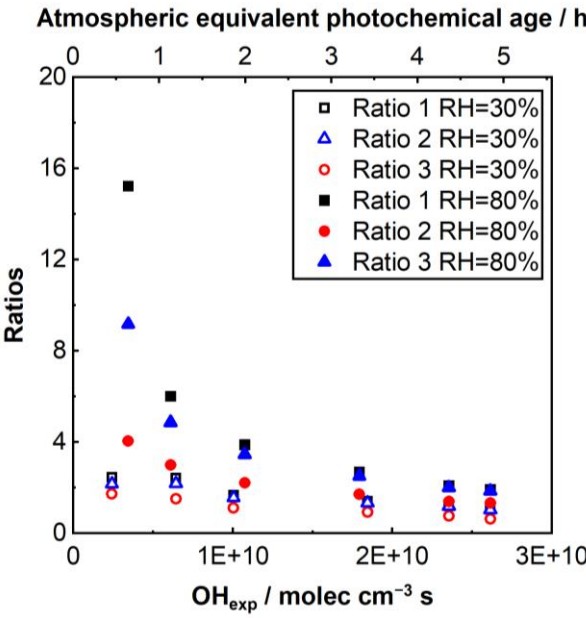


**Figure 6: Values of (Y'$_{MACR}$ + Y'$_{MVK}$) / Y'$_{HCHO}$ (Ratio 1), (Y'$_{MACR}$ + Y'$_{MVK}$) / (Y'$_{HCHO}$ + Y'$_{HMHP}$) (Ratio 2) and (Y'$_{MACR}$ + Y'$_{MVK}$) / (Y'$_{HCHO}$ + Y'$_{HMHP}$ + Y'$_{FA}$) (Ratio 3) under 30 % and 80 % RH.**

**3.3 Mechanisms of the multi-generation reactions**

As discussed above in Sect. 3.2, MACR, MVK, and β-ISOPOOH, as major first-generation photooxidation products of

isoprene via the HO$_2$ pathway, are of nearly equal importance in the presence of water vapor. Yet their further photooxidation pathways diverged. OH converts more than 70 % of β-ISOPOOH into β-IEPOX (Paulot et al., 2009; St Clair et al., 2016). Based on the rate constants of MACR-, MVK-, and β-IEPOX-derived RO$_2$ in cross-reactions with HO$_2$ and RO$_2$, and





intramolecular H-shift reactions abstracted from MCM v3.3.1, we estimated that in our experiments, 25 %, 25 %, and 50 % of MACR-derived $RO_2$ would process through the H-shift pathway, the $RO_2$ pathway, and the $HO_2$ pathway, respectively. β-

IEPOX-derived $RO_2$ would experience H-shift reactions and cross-reactions with $HO_2$ at a ratio of 1:2. For MVK-derived $RO_2$, reactions with $HO_2$ are still dominated. Given the distributions of the reaction pathways for MVK-, MACR-, and β-IEPOX-derived $RO_2$, we referred to the literature (Bates et al., 2014; Praske et al., 2015; Wennberg et al., 2018) and calculated the expected yields of a series of multi-generation products (including HCHO, MHP, PAA, MGLY, HACE, AA, and FA) under 30 % and 80 % RH, and the results are shown in Fig. 7.

According to Fig. 7, we found that the measured yields of most multi-generation fragmentation products exceed the expected ones, especially for AA and FA. The yield of AA was six times the expected. For FA, a yield of more than 0.2 cannot be explained by the mechanisms available in the literature. We suppose the increase in AA yield was partly due to the water effect on the $HO_2$ reaction of peroxyacetyl radical ($CH_3CO_3$). The $HO_2$ reaction of $CH_3CO_3$ produces AA with a yield of 0.25 in MCM v3.3.1. $CH_3CO_3$ complexes with $H_2O$ at a ratio of 0.2 % under 30 % RH and 0.5 % under 80 % RH (Clark et al., 2008).

The existence of water vapor possibly enhanced the branching ratio for AA in the $HO_2$ reaction of $CH_3CO_3$ via the formation of ternary hydrotetroxide intermediate like that put forward in the first-generation reactions. However, this mechanism can merely explain an increased AA yield of 3 % even if a 100 % AA yield was given to the $CH_3CO_3 + HO_2$ reaction, far from enough to explain the enormous AA yield in our experiments.

Here we tentatively attribute the enhancement in multi-generation FA and AA yields to the photo-aging process of SOA and

aqueous phase oxidation of oxygenated VOCs (OVOCs) in deliquesced particles. A previous study showed that the photodegradation of isoprene SOA under near-UV radiation (290−330 nm) releases FA and AA with a rate constant of $3.2 \times 10^{-6}$ and $6.5 \times 10^{-7}$ s$^{-1}$, respectively (Malecha and Nizkorodov, 2016). The aqueous-phase reactions may play a pivotal role under 80 % RH. The literature has provided evidence for the formation of FA and AA from the aqueous-phase photooxidation of dominant products from the $HO_2$ pathway. The aqueous-phase photodegradation of 2-methyltetrol, a major hydrolysis

product of IEPOX, is a strong source of FA and AA, with a yield as high as ~78 % for FA and ~53 % for AA (Cope et al., 2021). The aqueous-phase photooxidation of β-IEPOX produces AA at a yield of ~14 % (Otto et al., 2019). Moreover, small multi-functional carbonyls in isoprene SOA including HACE and MGLY are potential precursors for low-molecular-weight organic acids via dark- or photo-aging process in the aqueous phase (Liu et al., 2012). The appearance of unexpected high FA and AA concentrations is prevailing in laboratory simulations of gas-phase photooxidation of isoprene (Link et al., 2020) and

regions influenced by heavy biogenic emissions (Schobesberger et al., 2016; Alwe et al., 2019). Future studies may focus on illustrating the definite pathways for the formation of low-molecular-weight OVOCs including FA and AA in isoprene photooxidation.


(a) RH=30%

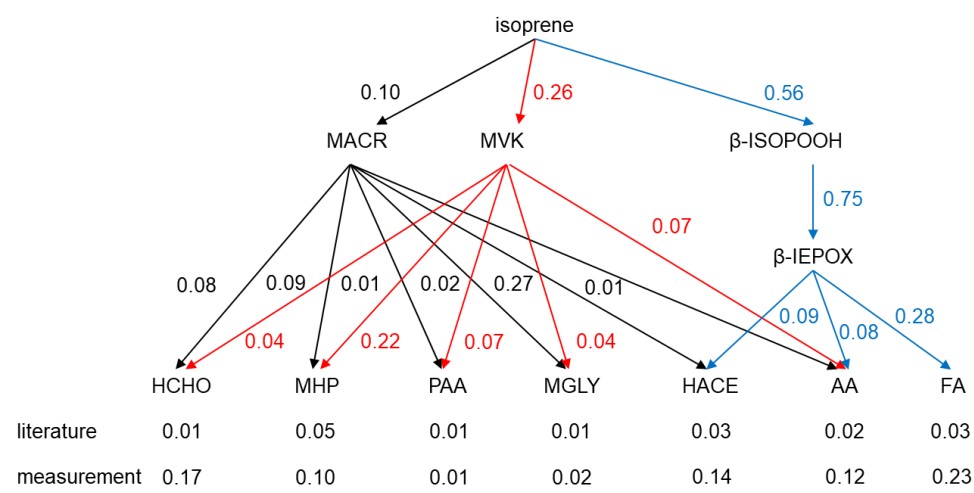

(b) RH=80%

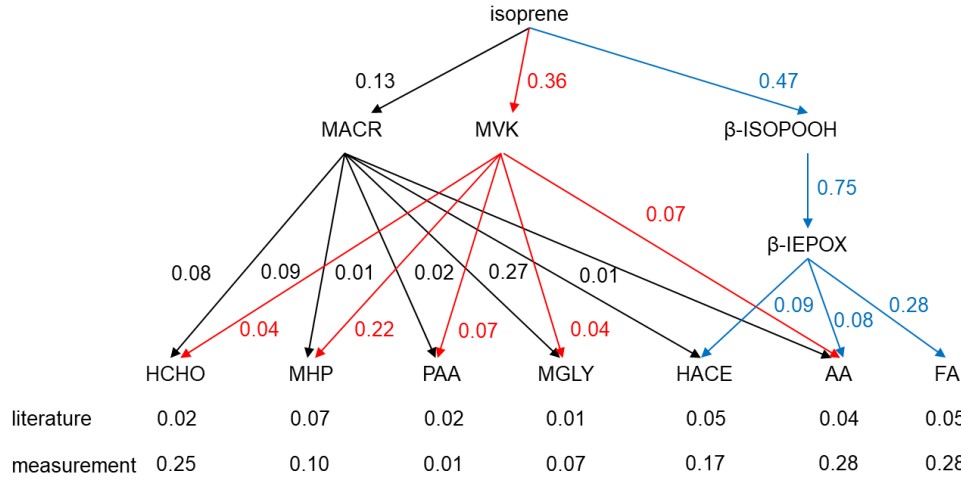

**Figure 7: Comparison of the yields of multi-generation products in literature and measurement under (a) 30% RH and (b) 80% RH**
**when atmospheric EPA equals ~15 h. Black arrows with values on the left side, MACR yield in this study and yields of MACR-derived products calculated based on Wennberg et al. (2018); red arrows with values on the right side, MVK yield in this study and yields of MVK-derived products calculated based on Praske et al. (2015); blue arrows with values on the right side, β-ISOPOOH yield in this study, β-IEPOX yield in St Clair et al. (2016), and yields of β-IEPOX-derived products under low NO conditions in Bates et al. (2014). The yields of HCHO and FA in the first-generation reactions were subtracted from their yields in measurement.**





### 3.4 Changes to the estimated emissions of the related products

Based on the updated branching ratios in the $HO_2$ reactions of β-ISOPOO in this study, the distribution of isoprene oxidation pathways, ISOPOO reaction pathways, and ISOPOO isomers in Bates and Jacob (2019), and the regional and global isoprene emissions in Sindelarova et al. (2014), we estimated the changes to β-ISOPOOH, MACR, and MVK emissions compared to those evaluated by a uniform branching ratio of 0.063 for MACR and MVK under 30 % and 80 % RH under four scenarios: East China, Southeast USA, Amazon, and global, with the first three corresponding to polluted, semi-polluted and remote areas, respectively. The RH falls in the range of 30−80 % in most parts of the world, except for those extremely arid and humid areas (Porter et al., 2021). Thus, we suggest that the values we obtained under 30 % RH represent the lower limit of the changes to their estimated emissions while those obtained under 80 % RH represent the upper limit. Table 2 listed the calculation results. Globally the estimated MACR and MVK emissions are enhanced by 4.68−12.4 and 18.2−34.1 Tg yr$^{-1}$, while the β-ISOPOOH emissions are suppressed by 38.6−78.3 Tg yr$^{-1}$. The changes are larger in the regions where biogenic emissions are dominated. A better interpretation of the relative importance of β-ISOPOOH and MACR helps understand the fate of isoprene in the atmosphere and improve the accuracy of the simulations of isoprene-derived SOA burdens and chemical compositions, as they are important SOA precursors under low-$NO_x$ and high-$NO_x$ conditions, respectively (Surratt et al., 2010; Lin et al., 2013; Krechmer et al., 2015; Nguyen et al., 2015).

**Table 2: Changes to estimated β-ISOPOOH, MACR, and MVK emissions (unit: Tg yr$^{-1}$) calculated based on the updated branching ratios in the cross-reactions of β-ISOPOO and $HO_2$ in this study under four scenarios.**

| Scenarios | RH = 30 % | | | RH = 80 % | | |
|---|---|---|---|---|---|---|
| | β-ISOPOOH | MACR | MVK | β-ISOPOOH | MACR | MVK |
| East China | −0.22 | 0.03 | 0.10 | −0.46 | 0.09 | 0.19 |
| Southeast USA | −1.45 | 0.19 | 0.67 | −2.96 | 0.50 | 1.26 |
| Amazon | −10.0 | 1.03 | 4.91 | −20.1 | 2.72 | 9.19 |
| Global | −38.6 | 4.68 | 18.2 | −78.3 | 12.4 | 34.1 |

Besides MACR, MVK, and β-ISOPOOH, a variety of low-molecular-weight OVOCs were also observed in our experiments, among which HMHP is the most special as a first-generation product. HMHP, as a prevailing organic hydroperoxide in the atmosphere (Hua et al., 2008; Qin et al., 2018), contributes to the regional sulfate burden considerably (Dovrou et al., 2022). HMHP is previously known as the product of gas-phase ozonolysis of alkene (Huang et al., 2013; Li et al., 2016; Gong et al., 2018) and in-cloud reactions of $H_2O_2$ and HCHO (Dovrou et al., 2022). Our study implies that isoprene photooxidation is also a potential source of HMHP in the atmosphere. Deep photooxidation of isoprene produces enormous amounts of FA and AA, especially under high RH. The FA yield measured in our experiments doubled that as a base case in the global modeling (~13%) when considering isoprene ozonolysis (Millet et al., 2015). Satellite sensors show an annual production of 100−120 Tg FA globally, three times that can be explained by known sources (Stavrakou et al., 2011). Given that isoprene oxidation contributes approximately 2/3 to simulated summertime tropospheric FA productions (Millet et al., 2015), a double in FA yield can lead

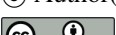



to an increase of 20−30 Tg yr$^{-1}$ in estimated FA emissions. Even so, the global FA productions are still underpredicted by nearly 40 %.

## 4 Conclusions

In this study, we investigated the first- and multi-generation gas-phase photooxidation regimes of isoprene via the HO$_2$ pathway. The experiments took place in an OFR with O$_3$ photolysis in the presence of water vapor as the OH source. We found that in the first-generation reactions, the branching ratios for MACR and MVK in β-ISOPOO + HO$_2$ reactions were enhanced by a factor of near three and more than five under 30 % and 80 % RH than those under dry conditions. We suggest that a water-induced change in the structures of the reaction intermediates from a hydrogen-bonded complex to a hydrotetroxide can explain

this phenomenon. We recommend taking the water-enhanced branching ratios for fragmentation into consideration in future modeling (the related mechanisms and kinetics are listed in Table S6). Some assumptions were made to achieve the above-mentioned branching ratios because the research on the water effect on the kinetics and mechanisms of the self/cross-reactions of RO$_2$ is limited. Thus, we call for more related experimental and theoretical studies, whose results benefit illuminating the fate of isoprene and its photooxidation products including but not limited to MACR, MVK, ISOPOOH, and IEPOX in the

ambient atmosphere.

We also discovered a promotion in the yields of fragmented multi-generation products when water vapor exists, especially those of FA and AA under high RH. This was tentatively attributed to the aqueous-phase photooxidation of products from the HO$_2$ pathway such as β-IEPOX and 2-methyltetrol in aerosol liquid water. However, despite a doubled yield observed in this study than the model baseline, there is still a 40 % gap between the observed and the modeled FA productions. FA accelerates

new particle formation (Zhang et al., 2012) and regulates precipitation and aerosol acidities (Vet et al., 2014). Future studies may focus on addressing the exact mechanisms for FA formation in isoprene photooxidation and evaluating the importance of various pathways in the ambient atmosphere to narrow the gap between modeled and measured acid emissions.

*Data availability*. The data are accessible by contacting the corresponding author (zmchen@pku.edu.cn).

*Supplement*. The supplement related to this article is available online at:

*Author contributions*. ZC and JX designed the research; JX performed the experiments, analysed the data, and wrote the manuscript draft; ZC, XQ, and PD reviewed and edited the manuscript.

*Competing interests*. The authors declare that they have no conflict of interests.



*Acknowledgements.* The authors gratefully thank the editor and referees for their constructive comments on the manuscript.

*Financial support.* This research was supported by the National Natural Science Foundation of China (grant no. 41975163) and the National Key Research and Development Program of China (grant no. 2016YFC0202704).

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
