# Peer review of "Water enhances the formation of fragmentation products via the cross-reactions of RO2 and HO2 in the photooxidation of isoprene"

_Atmospheric Chemistry and Physics, 2022_

## Referee Comment (RC2)

MS No. acp-2022-444

The authors are presenting experimental results on product formation from the OH + isoprene reaction conducted in a flow-through apparatus with a residence time of about 60 s applying very high OH radical concentrations. OH radicals were produced by means of ozone photolysis in the presence of water vapor. Total peroxides were determined by an iodometric method and MVK and MACR (and other carbonyls) by means of DNPH derivatization. No direct measurements of $RO_2$ radicals, $HO_2$ or hydroxy hydroperoxides are provided. Based on modeling results the authors concluded that $RO_2$ radicals mainly reacted with $HO_2$ or via $RO_2$ self- and cross-reactions. Nothing is said regarding the possible contribution of the $RO_2$ + NO reaction for product formation. The authors obviously neglected the $RO_2$ + OH reaction, even for the very high OH levels in the experiments. No explanation for that is given.

The authors are stating as a result of their experiments increasing formation yields of MVK and MACR from the $HO-C_5H_8O_2$ + $HO_2$ reaction with rising RH, i.e., an increase of $C_4$ carbonyl production by a factor of 5 increasing RH from "dry" to 80%. This result is very surprising and would change our understanding of product formation from OH + isoprene for low-NO conditions, as also pointed out in the manuscript based on modeling results.

I think there are some weak points in the experiment, especially in the analytical technique:

- Now it´s well-known that hydroxy hydroperoxides, which should be the primary product of $HO-C_5H_8O_2$ + $HO_2$ due to our current knowledge, are labile substances that tend to decompose at surfaces finally forming the corresponding carbonyls, see f.i. doi.org/10.1002/2014GL061919. This path of heterogeneous MVK/MACR formation was neglected, or checked the authors a contribution from that? Dosing the different $HO-C_5H_8OOH$ isomers should clearly show what happens during sampling and DNPH derivatization. This test is necessarily needed in order to trust the carbonyl yields. Note, the "questionable" RH dependence in MVK/MACR yields in former studies was most likely due to heterogeneous MVK/MACR production.
- Why running the experiments with super high OH levels? There is no need for OH exposures simulating a day. A possible water-mediated $HO-C_5H_8O_2$ + $HO_2$ reaction proceeds at a time scale of seconds or less for atmospheric reactant concentrations. So it would be much better to run the reaction for almost atmospheric conditions, pushing back unwanted pathways that can become important due to the high radical levels in the experiments.
- Nothing is said regarding hydrotrioxide formation in this reaction system. And also the hydrotrioxides are labile substances representing possible MVK/MACR precursors.

Although this work is dealing with a very interesting topic, I cannot recommend acceptance even after some modifications. I would like encourage the authors to reinvestigate the possible water-dependence in this reaction system using an improved experimental approach including the test for heterogeneous carbonyl formation.

---

## Author Comment (AC1)

**Response to the comments of Reviewer #1**

We thank Anonymous Referee #1 for the review. We have fully considered the comments and responded to these comments below in blue text. The revisions in the manuscript are presented in red text. Line numbers in our response correspond to those in the revised manuscript.

(Q=Question, A=Answer, C=Changes in the revised manuscript)

Xu et al. investigated the effects of water on the formation of fragmentation products in isoprene photooxidation with a series of oxidation flow reactor (OFR) experiments. They found that water enhanced MVK and MACR formation and proposed water-assisted mechanisms for the reactions of  $\beta$ -ISOPOO with HO2 to explain the observed fragmentation. I believe that the authors did the experiments carefully and reported a lot of useful details about them in the paper. I also think that the observations from the experiments are reliable. However, I do not agree with how the authors interpreted some key observations.

First, I find it highly implausible that HMHP formed via CH2OH (+O2) -> HOCH2OO (+HO2) -> HMHP in the gas phase. Theoretical calculations showed that CH2OH and O2 are too energetic for HOCH2OO to be stable. Even transient existence of HOCH2OO in this pathway will also end up with HCHO and HO2 in picoseconds (Dibble, 2002). While I agree that the formation of HMHP as a first-generation product likely involves some C1 fragment(s), I believe that condensed phase is needed for the ultrafast dissipation of energy excess of CH2OH+O2.

In the paper the authors have ruled out the reactor walls as this condensed phase. They suggested that isoprene-derived SOA may provide some aqueous phase volume. However, I do not think that isoprene-derived SOA would be enough. The SOA yield of isoprene is low even at equilibrium and without aerosol seed added a residence time of  $\sim$ 60 s is too short for SOA growth in OFR experiments (Palm et al., 2016).

The authors also reported much more formation of formic and acetic acids than explained by the mechanisms that the authors proposed. The strong production of FA and AA, together with the formation of HMHP as a first-generation product, lead me to think about a possible role that aqueous-phase chemistry could play in the experiments in this study.

I suspect that the movable sampling tube might have provided the aqueous phase needed. The Teflon lining might have adsorbed water (Huang et al., 2018) and its length and surface-to-volume ratio could be high enough to affect the experimental results.

I think that the authors should verify the possibility of aqueous-phase chemistry (not just in the sampling tube as I suspected) for the formation of HMHP, FA and AA. If they were not formed in the aqueous phase, more convincing mechanisms of their formation are needed for gas-phase water-assisted mechanisms for MVK and MACR formation to be more plausible.

A: Thank you for constructive comments. You are putting forward three questions in the major comments: (1) the possibility of aqueous-phase chemistry for the formation of HMHP, FA and AA; (2) gas-phase mechanisms for HMHP formation; (3) gas-phase mechanisms for FA and AA formation. In the following we will respond to the three questions separately.

Q (1): The possibility of aqueous-phase chemistry for the formation of HMHP, FA and AA.

A (1): We did control experiments to see if HMHP, FA, and AA are formed via aqueous-phase reactions. We suppose the aqueous-phase reactions possibly take place in the coiling tube, in the sampling tubes, or on the reactor walls given that isoprene-derived SOA is not the place.

The aqueous-phase reaction of  $H_2O_2$  and HCHO in the coiling tube is a possible source for HMHP. We estimated the amount of HMHP formed in the coiling tube based on the concentration of HCHO (calculated from its measured concentrations in the gas phase and Henry's law constant, 3242.4 M / atm) (Sander, 2015) and  $H_2O_2$  (measured by the HPLC) in the rinsing solution and the equilibrium constant of their reaction (172  $M^{-1}$ ) (Dovrou et al., 2022). Note that the value we estimated here refers to the upper limit because the time scale of HMHP equilibration is 30–60 min (Zhao et al., 2013), yet the residence time of the rinsing solution in the coiling tube is less than 10 min before measured by the HPLC. HMHP formed in the coiling tube consists of no more than 17 % of the detected HMHP, with an increasing proportion with  $OH_{exp}$ , and this part was subtracted from its yield shown in Fig. 2. The formation of AA and FA in the coiling tube was excluded because we analyzed the samples twice continuously (the time resolution for each measurement is 31 min) and the changes in FA and AA concentrations were within 3 %, indicating that the concentrations of FA and AA in the rinsing solution are stable within a time scale of tens of minutes.

Figure 2:  $OH_{exp}$ -dependent overall molar yields ( $Y'_{PR0,t}(t)$ ) of measured products at (a–d) 30 % RH and (e–h) 80 % RH in low and high  $OH_{exp}$  experiments. L, yields derived from low  $OH_{exp}$  experiments (Exp. 1 and 2); H, yields obtained from high  $OH_{exp}$  experiments (Exp. 3 and 4). The error bars represent ± standard deviation (± SD) based on 6 measurements.

To check the possibility of HMHP, FA, and AA formation in the sampling tubes, we doubled the length of sampling tube L1 (2.0 m, FEP, 1/8 in. o. d.) or L2 (3.4 m, FEP, 1/4 in. o. d.) (labeled in Fig. 1), and measured whether their concentrations changed. No obvious changes in HMHP (< 6 %), AA (< 6 %), and FA (< 4 %) were observed at

30 % and 80 % RH when the sampling gases passed through the additional length of the sampling tubes because of the extremely short residence time (0.22 s in L1 and 2.5 s in L2). The case on the OFR walls is more complicated because both photo and dark reactions may occur in the aqueous phase. We added another same OFR as we used in the experiments (OFR2) before sample collection to check the dark wall reactions. The concentrations of HMHP and FA decreased by 30 % and 5 %, respectively, due to the wall loss in OFR2, while the AA concentration increased by 10 %. We placed ~5 mL rinsing solution collected in Exp. 1 (RH=30 %) or Exp. 2 (RH=80 %) at an OHexp equivalent to  $2.77 \times 10^{10}$  molec cm-3 s into a quartz tube sealed at both ends and put it under the same 254 nm UV lamp as in the experiments for 60 s. We found the changes in FA and HMHP concentrations are restricted, while the AA concentration increased by 30 % after the irradiation. Consequently, we suggest the aqueous-phase chemistry on the reactor walls may have some influence on AA formation. However, its impact on FA and HMHP formation is minor.

---

## Author Comment (AC2)

**Response to the comments of Reviewer #2**

We thank Anonymous Referee #2 for the review of our manuscript. We have fully considered the comments and responded to these comments below in blue text. The revisions in the manuscript are presented in red text. Line numbers in our response correspond to those in the revised manuscript.

(Q=Question, A=answer, C=Changes in the revised manuscript)

The authors are presenting experimental results on product formation from the OH + isoprene reaction conducted in a flow-through apparatus with a residence time of about 60 s applying very high OH radical concentrations. OH radicals were produced by means of ozone photolysis in the presence of water vapor. Total peroxides were determined by an iodometric method and MVK and MACR (and other carbonyls) by means of DNPH derivatization. No direct measurements of RO2 radicals, HO2 or hydroxy hydroperoxides are provided. Based on modeling results the authors concluded that RO2 radicals mainly reacted with HO2 or via RO2 self- and cross-reactions. Nothing is said regarding the possible contribution of the RO2 + NO reaction for product formation. The authors obviously neglected the RO2 + OH reaction, even for the very high OH levels in the experiments. No explanation for that is given.

A: We are sorry that the direct measurements of RO2 and HO2 radicals and hydroxy hydroperoxides are beyond achievable based on our current analytical techniques. NO as an impurity in the cylinder gases (<0.2 ppbv) would react rapidly with the major oxidants in the OFR including OH, O3, and HO2, and be converted into NO2 in less than 1 s (Peng and Jimenez, 2020). Thus, the contribution of RO2 + NO reaction for MACR and MVK formation is negligible. This has been added to the revised manuscript. In the revised manuscript we have evaluated the contribution of RO2 + OH reaction to MACR and MVK formation. The OH pathway contributes < 3 % to the detected MACR and MVK. Please refer to our response to Q3 for details.

C: (Sect. 3.3 Line 254–256)

The contribution of ISOPOO + NO reaction for MACR and MVK formation is

negligible because NO as a possible impurity in the cylinder gases (< 0.2 ppbv) would be converted into NO2 by the major oxidants (OH, HO2, and O3) in the OFR in less than 1 s (Peng and Jimenez, 2020).

The authors are stating as a result of their experiments increasing formation yields of MVK and MACR from the HO-C5H8O2 + HO2 reaction with rising RH, i.e., an increase of C4 carbonyl production by a factor of 5 increasing RH from "dry" to 80%. This result is very surprising and would change our understanding of product formation from OH + isoprene for low-NO conditions, as also pointed out in the manuscript based on modeling results.

I think there are some weak points in the experiment, especially in the analytical technique:

Q1: Now it's well-known that hydroxy hydroperoxides, which should be the primary product of HO-C5H8O2 + HO2 due to our current knowledge, are labile substances that tend to decompose at surfaces finally forming the corresponding carbonyls, see f.i. doi.org/10.1002/2014GL061919. This path of heterogeneous MVK/MACR formation was neglected, or checked the authors a contribution from that? Dosing the different HO-C5H8OOH isomers should clearly show what happens during sampling and DNPH derivatization. This test is necessarily needed in order to trust the carbonyl yields. Note, the "questionable" RH dependence in MVK/MACR production.

A1: Thank you for your suggestion. We are sorry that  $\beta$ -1,2-ISOPOOH and  $\beta$ -4,3-ISOPOOH standards are currently not available. However, we carried out a series of control experiments to estimate the amount of heterogeneous MACR and MVK production in the sampling tubes by doubling their length, and on the OFR walls by introducing another same OFR before sample collection. The changes of MACR, MVK, HCHO, and C≥3 PO concentrations in different scenarios were measured simultaneously. We also evaluated the influence of DNPH derivatization on ISOPOOH isomers based on our unpublished data. We conclude from the results that the heterogeneous formation of MACR and MVK in the sampling tubes and during DNPH derivatization is minor, while that in the OFR is major. Please refer to the following for the details.

In Exp. 1 (RH=30 %) and Exp. 2 (RH=80 %) at the outlet of the OFR when OHexp is equivalent to  $2.77 \times 10^{10}$  molec cm-3 s, we doubled the length of sampling tube L1 (2.0 m, FEP, 1/8 inch o. d., labeled in Fig. 1), L2 (3.4 m, FEP, 1/4 inch o. d.), or L3 (1.0 m, FEP, 1/4 inch o. d.), or added another same OFR (OFR2) before carbonyls or peroxides sample collection, and measured how much the observed MACR, MVK, HCHO, and C $\geq$ 3 PO concentrations changed. The formation and loss of MACR and MVK from ozonolysis in the sampling tubes and OFR2 are negligible (< 0.01 ppbv). We found the changes in carbonyls concentrations were less than 4 % when the sampling gases passed through the additional length of L1, L2 and L3, within the uncertainty ranges (< 5 %). The changes in C $\geq$ 3 PO concentrations were less than 4% at 30 % RH and less than 7 % at 80 % RH, not obvious as well regarding the uncertainty ranges (< 10 %). However, non-negligible formation of MACR and MVK and loss of C $\geq$ 3 peroxides were noticed when the sampling gases passed through OFR2 at both 30 % and 80 % RH. The carbonyls were enhanced by 8 %, and the C≥3 peroxides decreased by 14 % at 30 % RH and 31 % at 80 % RH. Consequently, corrections are needed for heterogeneous carbonyl formation in the OFR. We define the loss fraction of ISOPOOH isomers in the OFR as  $LF_{OFR,4,3-ISOPOOH}$  and  $LF_{OFR,1,2-ISOPOOH}$ , and they can be calculated as Eq. 1 and 2:

$$LF_{OFR,4,3-ISOPOOH} = \frac{\Delta MACR_{OFR2}}{[4,3-ISOPOOH_{OFR2,IN}]}$$
(1)

$$LF_{OFR,1,2-ISOPOOH} = \frac{\Delta MVK_{OFR2}}{[1,2-ISOPOOH_{OFR2,IN}]}$$
(2)

[4,3 – *ISOPOOH*OFR2,IN] or [1,2 – *ISOPOOH*OFR2,IN] refer to the concentration of  $\beta$ -4,3-ISOPOOH or  $\beta$ -1,2-ISOPOOH at the inlet of OFR2. They are determined by multiplying the observed concentrations of C≥3 peroxides (9.2 ± 0.7 ppbv for Exp. 1 and 5.2 ± 0.4 ppbv for Exp. 2 at an OHexp equivalent to 2.77 × 1010 molec cm-3 s) and the modeled weight of the ISOPOOH isomers (83 % for  $\beta$ -1,2-ISOPOOH and 16 % for  $\beta$ -4,3-ISOPOOH) given the loss of ISOPOOH isomers is trivial in the sampling tubes.  $\Delta MACR_{OFR2}$  and  $\Delta MVK_{OFR2}$  refer to the changes in MACR and MVK concentrations in OFR2. The results of *LF*OFR at 30 % and 80 %

RH are presented in Table 2. It is noted that the  $LF_{OFR}$  in Table 2 refer to the LF of ISOPOOH isomers when the sampling gases pass through the entire OFR. The  $LF_{OFR}$  at different residence times derive from the pseudo-first-order reaction kinetic equation. Our results show that  $\beta$ -4,3-ISOPOOH is more labile than  $\beta$ -1,2-ISOPOOH, and their decomposition is positively related to RH, which is consistent with St Clair et al. (2016b). The wall loss fractions of H2O2 and a series of  $\alpha$ -hydroxyalkyl-hydroperoxides ( $\alpha$ -HHs) and peroxy acids were measured in the same OFR in our previous study (Huang et al., 2013). We found that the wall loss fraction of  $\beta$ -1,2-ISOPOOH is almost equivalent to that of HMHP, while that of  $\beta$ -4,3-ISOPOOH is twice that of HMHP.

The influence of DNPH derivatization on ISOPOOH isomers was evaluated based on our unpublished data obtained from previous isoprene + OH experiments under dry conditions. The concentration of isoprene was 600 ppbv. The photolysis of 7 ppmv H2O2 at 312 nm UV irradiation served as the OH source. The residence time in the OFR was 137 s. The consumed isoprene was  $176 \pm 8$  ppbv. Carbonyls were collected with DNPH cartridges and analyzed with the same method as in our experiments. The reported MACR and MVK yields were  $0.18 \pm 0.01$  and  $0.21 \pm 0.02$ . A box model was applied to estimate the supposed MACR and MVK yields in those experiments considering a 0.063 branching ratio for ISOPO in the HO2 reaction of ISOPOO and a wall loss rate constant of  $1.19 \times 10^{-3}$  for 1,2-ISOPOOH and  $3.66 \times 10^{-3}$  for 4,3-ISOPOOH (derived from  $LF_{OFR}$  obtained at 30 % RH). We suggest DNPH decomposes the ISOPOOH isomers if the MACR and MVK yields in the experiments are higher than the modeled values. The modeled yield is 0.19 for MVK and 0.18 for MACR, meeting with the observed one, indicating the loss of ISOPOOH isomers during DNPH derivatization is minor.

The setup for control experiments is added as Sect. 2.3 Control experiments. The calculation of LF as well as the discussion on the influence of DNPH derivatization on ISOPOOH isomers are added as Sect. 3.1 Corrections for heterogeneous carbonyl formation in the revised manuscript.

The losses of ISOPOOH isomers on the OFR walls were considered in the data analysis. Artifacts from heterogeneous carbonyl formation account for 5-12 %

(15–22 %) and 1–9 % (5–8 %) of the observed MACR and MVK at 30 % (80 %) RH, varied with the residence time in the OFR. They were subtracted from the observed concentrations. We have corrected the yields of MACR, MVK, HCHO and C $\geq$ 3 peroxides in Fig. 2 and the MACR and MVK concentrations in Fig. 5. All calculations involving MACR and MVK concentrations have been reprocessed. Please refer to the following for the major revisions in the text.

C1: (Sect. 2.3 Line 131–138)

**2.3 Control experiments**

β-ISOPOOH isomers are labile substances that tend to decompose at surfaces, forming the corresponding carbonyls MACR and MVK (Rivera-Rios et al., 2014), and HCHO as a by-product. Control experiments were conducted to evaluate the heterogeneous carbonyl formation in the OFR and sampling tubes. In Exp. 1 (RH=30 %) and Exp. 2 (RH=80 %) at an OHexp equivalent to  $2.77 \times 10^{10}$  molec cm-3 s, we doubled the length of sampling tube L1 (2.0 m, FEP, 1/8 in. o. d., labeled in Fig. 1), L2 (3.4 m, FEP, 1/4 in. o. d.), or L3 (1.0 m, FEP, 1/4 in. o. d.), or added another same OFR (OFR2) before carbonyls or peroxides sample collection, and measured how much the observed MACR, MVK, HCHO, and C≥3 PO concentrations changed. See Sect. 3.1 for the results and discussion.

(Sect. 3.1 Line 153–191)

**3.1 Corrections for heterogeneous carbonyl formation**

The changes in MACR, MVK, HCHO, and C $\geq$ 3 PO were measured when additional L1, L2, L3, or OFR2 presented. The formation and loss of MACR and MVK from ozonolysis in the sampling tubes and OFR2 are negligible (< 0.01 ppbv). We found the changes in carbonyls concentrations were less than 4 % when the sampling gases passed through the additional length of L1, L2 and L3, within the uncertainty ranges (< 5 %). The changes in C $\geq$ 3 PO concentrations were less than 4% at 30 % RH and less than 7 % at 80 % RH, not obvious as well regarding the uncertainty ranges (< 10 %). However, non-negligible formation of MACR and MVK and loss of C $\geq$ 3 peroxides were noticed when the sampling gases passed through OFR2 at both 30 % and 80 % RH. The carbonyls were enhanced by 8 %, and the C $\geq$ 3 peroxides decreased by 14 % at 30 % RH and 31 % at 80 % RH. Consequently, corrections are needed for heterogeneous carbonyl formation in the OFR. We define the loss fraction of ISOPOOH isomers in the OFR as  $LF_{OFR,4,3-ISOPOOH}$  and  $LF_{OFR,1,2-ISOPOOH}$ , and they can be calculated as Eq. 1 and 2:

$$LF_{OFR,4,3-ISOPOOH} = \frac{\Delta MACR_{OFR2}}{[4,3-ISOPOOH_{OFR2,IN}]}$$
(1)

$$LF_{OFR,1,2-ISOPOOH} = \frac{\Delta MVK_{OFR2}}{[1,2-ISOPOOH_{OFR2,IN}]}$$
(2)

$$[4,3 - ISOPOOH_{OFR2,IN}]$$
 or  $[1,2 - ISOPOOH_{OFR2,IN}]$  refer to the

concentration of  $\beta$ -4,3-ISOPOOH or  $\beta$ -1,2-ISOPOOH at the inlet of OFR2. They are determined by multiplying the observed concentrations of C $\geq$ 3 peroxides (9.2 ± 0.7 ppbv for Exp. 1 and 5.2  $\pm$  0.4 ppbv for Exp. 2 at an  $OH_{exp}$  equivalent to 2.77  $\times$   $10^{10}$ molec cm-3 s) and the modeled weight of the ISOPOOH isomers (83 % for  $\beta$ -1,2-ISOPOOH and 16 % for  $\beta$ -4,3-ISOPOOH) given the loss of ISOPOOH isomers is trivial in the sampling tubes.  $\Delta MACR_{OFR2}$  and  $\Delta MVK_{OFR2}$  refer to the changes in MACR and MVK concentrations in OFR2. LF0FR at 30 % and 80 % RH are presented in Table 2. It is noted that the  $LF_{OFR}$  in Table 2 refers to those values when the sampling gases pass through the entire OFR. The  $LF_{OFR}$  at different residence times derive from the pseudo-first-order reaction kinetic equation. Our results show that  $\beta$ -4,3-ISOPOOH is more labile than  $\beta$ -1,2-ISOPOOH, and their decomposition is positively related to RH, which is consistent with St Clair et al. (2016b). The wall loss fractions of  $H_2O_2$  and a series of  $\alpha$ -hydroxyalkyl-hydroperoxides ( $\alpha$ -HHs) and peroxy acids were measured in the same OFR in our previous study (Huang et al., 2013). We found that the wall loss fraction of  $\beta$ -1,2-ISOPOOH is almost equivalent to that of HMHP, while that of  $\beta$ -4,3-ISOPOOH is twice that of HMHP.

The influence of DNPH derivatization on ISOPOOH isomers was evaluated based on our unpublished data obtained from previous isoprene + OH experiments under dry conditions (600 ppbv isoprene + 7 ppmv H2O2 + 312 nm UV irradiation). Carbonyls were collected with DNPH cartridges and analysed with the same method as in our experiments. The reported MACR and MVK yields were  $0.18 \pm 0.01$  and  $0.21 \pm 0.02$ . A box model was applied to estimate the supposed MACR and MVK yields in those experiments considering a 0.063 branching ratio for MACR or MVK formation in the HO2 reaction of ISOPOO and a wall loss rate constant of  $1.19 \times 10^{-3}$  for  $\beta$ -1,2-ISOPOOH and  $3.66 \times 10^{-3}$  for  $\beta$ -4,3-ISOPOOH (derived from  $LF_{OFR}$  obtained at 30 % RH). We suggest DNPH decomposes the ISOPOOH isomers if the MACR and MVK yields in the experiments are higher than the modeled values. However, their modeled yields (0.19 for MVK and 0.18 for MACR) meet with the observed ones, indicating the loss of ISOPOOH isomers during DNPH derivatization is minor.

From the above discussions, we conclude that the heterogeneous formation of MACR and MVK in the sampling tubes and during DNPH derivatization is minor, while that in the OFR is major. The losses of ISOPOOH isomers on the OFR walls were considered in the data analysis. Artifacts from heterogeneous carbonyl formation account for 5-12 % (15-22 %) and 1-9 % (5-8 %) of the observed MACR and MVK at 30 % (80 %) RH, varied with the residence time in the OFR. They were subtracted from the observed concentrations.

Table 2: Loss fractions of  $\beta$ -4,3-ISOPOOH and  $\beta$ -1,2-ISOPOOH in the OFR (*LF**OFR*) at 30 % (Exp. 1) and 80 % RH (Exp. 2).

| LF OFR /% | RH=30 %    | RH=80 %     |
|----------------------|------------|-------------|
| β-1,2-ISOPOOH        | $7 \pm 11$ | $23 \pm 12$ |
| β-4,3-ISOPOOH        | $20\pm 8$  | $54\pm14$   |

(Sect. 3.2 Line 206–208)

The average yields of MACR and MVK were  $10.4 \pm 4.5$  % and  $20.1 \pm 5.7$  % at 30 % RH (Exp. 1) and  $15.4 \pm 3.3$  % and  $34.1 \pm 5.8$  % at 80 % RH (Exp. 2).

(Sect. 3.3 Line 270–273)

The yield of MACR from the reaction of  $\beta$  4-OH, 3-OO ISOPOO radical ( $\beta$ -4,3-ISOPOO) with HO2 is 2.3 ± 4.5 % at 30 % RH and 4.3 ± 3.3 % at 80% RH, while the yield of MVK from the reaction of  $\beta$  1-OH, 2-OO ISOPOO ( $\beta$ -1,2-ISOPOO) with HO2 is 10.1 ± 5.7 % at 30 % RH and 16.8 ± 5.8 % at 80 % RH.

(Sect. 3.3 Line 274–276)

The yield of  $\beta$  1-OH, 2-OOH ISOPOOH ( $\beta$ -1,2-ISOPOOH) and  $\beta$  4-OH, 3-OOH

ISOPOO ( $\beta$ -4,3-ISOPOOH) was 47.1 ± 9.7 % and 21.0 ± 9.7 % at 30 % RH, and 33.9 ± 8.1 % and 18.6 ± 8.1 % at 80 % RH.

(Sect. 3.3 Line 285–287)

The branching ratio for MACR in the HO2 reaction of  $\beta$ -4,3-ISOPOO is 0.129 ± 0.080 at 30 % RH and 0.141 ± 0.097 at 80 % RH, while that for MVK in the HO2 reaction of  $\beta$ -1,2-ISOPOO is 0.230 ± 0.070 at 30 % RH and 0.345 ± 0.103 at 80 % RH.

Figure 1: Overview of the experimental apparatus.